# Estrogen Receptors in Epithelial-Mesenchymal Transition of Prostate Cancer

**DOI:** 10.3390/cancers11101418

**Published:** 2019-09-23

**Authors:** Erika Di Zazzo, Giovanni Galasso, Pia Giovannelli, Marzia Di Donato, Antonio Bilancio, Bruno Perillo, Antonio A. Sinisi, Antimo Migliaccio, Gabriella Castoria

**Affiliations:** 1Dipartimento di Medicina di Precisione, Università degli Studi della Campania “Luigi Vanvitelli”, 80138 Naples, Italy; erika.dizazzo@unimol.it (E.D.Z.); giovanni.galasso@unicampania.it (G.G.); pia.giovannelli@unicampania.it (P.G.); marzia.didonato@unicampania.it (M.D.D.); antonio.bilancio@unicampania.it (A.B.); 2Dipartimento di Medicina e Scienze della Salute “V. Tiberio”, Università degli Studi del Molise, 86100 Campobasso, Italy; 3Istituto di Scienze dell’Alimentazione—Consiglio Nazionale delle Ricerche, 83100 Avellino, Italy; perillo@unina.it; 4Dipartimento di Scienze Mediche, Chirurgiche, Neurologiche, Metaboliche e dell’Invecchiamento, Università degli Studi della Campania “Luigi Vanvitelli”, 80138 Naples, Italy; antonio.sinisi@unicampania.it

**Keywords:** estrogens, estrogen receptors (ERs), prostate cancer (PC), epithelial mesenchymal transition (EMT), metastasis

## Abstract

Prostate cancer (PC) remains a widespread malignancy in men. Since the androgen/androgen receptor (AR) axis is associated with the pathogenesis of prostate cancer, suppression of AR-dependent signaling by androgen deprivation therapy (ADT) still represents the primary intervention for this disease. Despite the initial response, prostate cancer frequently develops resistance to ADT and progresses. As such, the disease becomes metastatic and few therapeutic options are available at this stage. Although the majority of studies are focused on the role of AR signaling, compelling evidence has shown that estrogens and their receptors control prostate cancer initiation and progression through a still debated mechanism. Epithelial versus mesenchymal transition (EMT) is involved in metastatic spread as well as drug-resistance of human cancers, and many studies on the role of this process in prostate cancer progression have been reported. We discuss here the findings on the role of estrogen/estrogen receptor (ER) axis in epithelial versus mesenchymal transition of prostate cancer cells. The pending questions concerning this issue are presented, together with the impact of the available data in clinical management of prostate cancer patients.

## 1. Introduction

Despite diagnostic, clinical and therapeutic efforts, prostate cancer (PC) still represents a major urological disease associated with a significant morbidity. This cancer represents, indeed, the most common malignancy and the second most common cause of cancer death among men in Western society [1,2]. Significant advances have been made in diagnostic tools and therapeutic approaches of PC. They are based on the use of new radiotracers for imaging and patient follow-up [3,4,5]. Furthermore, new potent androgen synthesis inhibitors or androgen receptor (AR) antagonists, such as abiraterone and enzalutamide, have improved the survival rate in PC patients [6,7]. Nevertheless, PC often develops resistance to these therapeutics and progresses to castration-resistant PC (CRPC) stage, which is characterized by adaptation to castrate condition and invasiveness. Metastatic spread of CRPC cells still represents the major cause of PC-related death and novel compounds, such as taxanes, poly (adenosine diphosphate-ribose) polymerase, and programmed cell death 1 (PD-1) inhibitors, have entered clinical trials [8]. However, a precise understanding of the molecular basis for cell spreading and drug-resistance patterns in PC still remains a challenge.

Epithelial-mesenchymal transition (EMT) is a highly conserved evolutionary cellular process that allows immotile epithelial cells to convert into mesenchymal cells, endowed with a migratory phenotype. The relevance of the EMT process was initially recognized during the critical stages of embryonic morphogenesis, where EMT plays a crucial role in tissue remodeling and development. Such a process can be recapitulated in human diseases, including cancer [9,10,11,12]. As such, transformed cells switch from an epithelial to mesenchymal-like phenotype, while responding to intrinsic genetic and molecular alterations, or extrinsic microenvironment stimuli. This switch leads to invasion into surrounding stroma and then vasculature to ultimately induce colonization at a distant pre-metastatic niche [9]. Since this mechanism would enable the invasive properties of cancer cells, its inhibition might offer new promising opportunities to restrain the metastatic spreading of transformed cells and the related mortality. 

While it is largely accepted that EMT is involved in spreading, stemness, and drug-resistance of PC cells, conflict still remains about the role of steroids and sex steroid receptors (SRs) in this process. We present here the findings so far obtained on the molecular events underlying the EMT controlled by the estrogen/estrogen receptors (ERs) axis in different models of prostate tissues and PC. The impact of these findings in clinical management of PC patients will be also discussed.

## 2. Epithelial-Mesenchymal Transition (EMT) in Prostate Cancer (PC)

Epithelial cells are normally held together in a single layer through cell-cell interactions mediated by cadherins, mainly E-cadherin. Reduction in E-cadherin expression levels and other epithelial markers, such as β-catenin and occludin, together with the gain of mesenchymal markers (e.g., N-cadherin, Zeb-1/2, vimentin, Snail, and Twist) is a feature of EMT that is observed in more aggressive tumors, resulting in the decrease of intercellular adhesion, loss of epithelial cell polarity, de-differentiation into an amorphous cell, and increased motility [13,14]. 

Reduced E-cadherin expression was detected many years ago in specimens from PC patients with aggressive disease. However, its association with the profile related to EMT and the patient’s outcome were not investigated [15,16]. It was subsequently shown that increased N-cadherin and decreased E-cadherin expression levels positively correlate with poor prognosis and disease progression in PC patients [17]. Findings in cultured cells from aggressive PC confirmed the clinical evidence [18,19,20]. Despite being intriguing, these results were almost exclusively obtained from in vitro PC cultured cells or human cancer specimens. Nonetheless, they opened new frontiers in targeted therapies, as N-cadherin was thought to be a potential therapeutic target in human cancers [21,22]. Further studies showed that EMT promotes a castration-resistant PC (CRPC) state in various PC models and monoclonal antibodies targeting N-cadherin reverses such transition [23]. Moreover, Sun and colleagues reported that androgen deprivation therapy (ADT), frequently used as the first-line treatment in PC, induces EMT in normal prostate as well as in PC cells [24]. Both these reports offered valuable insights into the therapeutic approach of PC.

The main players responsible for EMT were then dissected in PC models as well as prostate cancer-associated fibroblasts (CAFs). Studies in preclinical models of PC showed that activation of both phosphoinositide 3-kinase (PI3-K)/AKT and Ras/mitogen-activated protein kinase (MAPK) pathways allows EMT and metastatic spreading [25,26]. It is noteworthy that both of these pathways can be rapidly activated by estradiol stimulation in breast cancer (BC)- and PC-derived cells [27,28,29]. Thus, PC cells might undergo EMT because of the hormonal activation of these signaling pathways. 

It was also reported that the tumor microenvironment as well as changes in DNA methylation of CAFs promote acquisition of stemness traits, EMT and metastatic spreading in PC cells [30,31]. We have consistently observed that prostate CAFs from human specimens promote EMT in a 3D model of PC cells (unpublished results). Lastly, the role of EMT in therapy escape was highlighted by clinical evidence indicating that EMT drives docetaxol-resistance and promotes the risk of relapse in PC patients [32]. 

Despite these interesting findings, the direct role for EMT in PC progression, dissemination of circulating tumor cells (CTCs) and seeding of metastases still remains to be completely elucidated, likely because of the lack of models recapitulating the complexity of metastatic processes in vivo. By using a model that allows the dynamic tracking of EMT process in vivo, mesenchymal-like tumor cells that have fully completed the EMT program, as well as EMT tumor cells, which are in a transitory state between epithelial and mesenchymal programs, have been isolated and characterized from a mouse PC model. Collected results have shown that only EMT tumor cells have the capacity to complete the invasion-metastasis cascade, since they harbor the plasticity to readily transit between epithelial and mesenchymal states [33]. These findings are of value in the therapeutic approach of PC, since they support the use of drugs targeting PC cell plasticity, rather than therapeutics specifically targeting the epithelial or mesenchymal state. Targeting of mesenchymal-like tumor cells and the mesenchymal state could be, indeed, useful in preventing PC dissemination. However, by promoting the reversion to an epithelial state (mesenchymal versus epithelial transition, MET), such strategies might paradoxically fuel metastatic growth. 

Nowadays, it is largely recognized that EMT is a dynamic process allowing epithelial cancer cells to acquire stemness as well as invasive properties and evade senescence, apoptosis, and anoikis, which are needed for tumor dissemination and metastasis [12,34]. Findings obtained in cultured cells as well as genetically engineered murine PC models and human PC specimens support the conclusion that PC often exhibits an EMT-like state, characterized by changes in the expression of different markers, such as E-cadherin, vimentin, and N-cadherin. 

Different regulators, such as inducers, controllers and effectors of the EMT program have been identified in PC. To date, the most widely accepted view proposes that tumor microenvironment provides local signals that promote EMT program. Among them, growth factors, such as the epidermal growth factor (EGF), fibroblast growth factor (FGF), insulin-like growth factor 1 (IGF-1), hepatocyte-derived growth factor (HGF), and platelet-derived growth factor (PDGF) trigger activation of phosphoinositide 3-kinase (PI3-K) and Ras signaling pathways. They, in turn, activate the downstream effectors mitogen-activated protein kinase (MAPK), glycogen synthase kinase-3β (GSK3) and nuclear factor kappa-light-chain-enhancer of activated B cells (NF-kB), which increase the activity of Snail and Twist, thereby inducing the expression of mesenchymal proteins [reference 35 and references therein]. Since the ligand-activated steroid receptors (SRs), including estrogen receptors (ERs) and androgen receptor (AR), rapidly activate Src or PI3-K and their downstream pathways in PC and BC-derived cells [29], it could be argued that estrogens and androgens promote EMT through activation of extranuclear signaling pathways in these cancers. 

Among growth factors (GF), transforming growth factor β (TGF-β) deserves particular interest, since it plays a major role in physiological EMT. TGF-β might promote EMT through the SMAD (small mothers against decapentaplegic homologs)- independent or -dependent signaling. Accordingly, TGF-β binds to its type II and type III receptors (TGF-βRII and TGF-βRIII), thereby leading to recruitment and phosphorylation of the type I receptor (TGF-βRI). Activation of PI3-K and Ras then follows, and expression of EMT-inducing proteins increases (SMAD-independent signaling). The SMAD-dependent signaling, instead, proposes that phosphorylation of TGF-βRI by the Ser/Thr kinase activity of TGF-βRII creates a docking site in the Gly/Ser–rich domain of TGF-βRI, which recruits the transcription factors SMAD2 and SMAD3. SMADs are consequently phosphorylated at Ser residues in their C-terminal domain, facilitating the formation of a complex with the co-activator SMAD4. This signal allows their subsequent translocation into the nucleus and activation of zinc finger E-box-binding homeobox (ZEB), which finally leads to expression of mesenchymal markers [9,10,11,12,13,14,35]. 

Other pathways, however, control EMT in PC. Stable overexpression of the secreted Wingless-INT (Wnt) antagonists secreted frizzled-related proteins (sFRPs) increases the expression of epithelial markers, reduces invasiveness and simultaneously downregulates Snail Family Transcriptional Repressor 2 (SNAI2) and Twist-related protein 1 (TWIST1) in PC cells. This is consistent with the idea that canonical Wnt is implied in EMT of PC through activation of GSK3β. Stabilization, nuclear translocation of β-catenin, as well as phosphorylation of SNAI1 then follow. This latter event promotes SNAI1 nuclear localization and enables its transcriptional activity [35] (and references therein). Again, the Notch signaling converges with TGF-β and Wnt signaling at the induction of direct transcriptional repressors of E-cadherin, including Snail/Slug, Twist, and ZEB1/2 [13]. Four different Notch transmembrane receptors can be activated by membrane-tethered ligands in mammals. Such activation initiates a proteolytic cleavage of Notch receptors, with the subsequent release of the Notch intracellular domain (NICD), which then translocates into the nucleus and controls expression of genes involved in EMT, such as Snail. As discussed later, the Notch pathway is emerging as a candidate for EMT and metastatic events in PC cells, since estrogens control this pathway through ERα activation in these cells. Figure 1 depicts the most important pathways involved in EMT of PC. 

Altogether, these findings support the concept that molecules underlying EMT might represent new PC biomarkers and ‘druggable’ targets for the development of novel therapeutics. The degree to which the EMT state results from activation of estrogen/ER axis in prostate tissue and PC is, however, still debated. The subsequent sections in this manuscript aim to address this issue. 

## 3. Estrogen Receptors (ERs) in Prostate Cancer (PC)

Normal human prostate expresses both the ER isoforms, α and β, which mediate the estrogen effects in these cells. ERα is predominantly localized in prostatic stroma. As such, its effect on epithelial cells can be considered to be indirect [37]. In contrast, ERβ, which was identified and cloned later [38,39,40], is expressed in the epithelial compartment of the gland, where it regulates epithelial proliferation and differentiation [41]. 

It is largely accepted that ERα mediates the adverse effects (i.e., survival, proliferation, and inflammation) induced by estrogens, while ERβ mediates the protective and anti-apoptotic estrogen effects in PC [41,42]. Thus, ERα may be regarded as an oncogene, while ERβ may elicit anti-tumor activity in PC. As such, it might be considered an onco-suppressor gene. Many findings, including ours, have questioned this latter concept. ERβ drives S-phase entry and proliferation in LNCaP cells challenged with estrogens or androgens [27] or epidermal growth factor (EGF) [43]. Activation of β-catenin mediated by ERβ increases cyclin D1 expression and proliferation in PC3 cells [44]. Again, ERβ but not ERα promotes survival and migration in the CPEC cell line established from PC patients, as shown by using the specific ER agonists, 2,3-bis(4-Hydroxyphenyl)-propionitrile (DPN) and 4,4′,4″-(4-Propyl-[1H]-pyrazole-1,3,5-triyl)trisphenol (PPT) [45]. Further, the two splice variants of ERβ, 2 and 5, are associated with poor PC prognosis and promote migration and invasiveness in PC cells [46]. Altogether, these findings support an oncogenic role for ERβ. 

Given these results, it might be argued that experimental pitfalls due to the type of ERβ antibodies used are responsible for the conflicting findings so far reported in the literature. Validation of commonly used and commercially available ERβ antibodies has demonstrated that some of them either detect ERβ in specific experimental conditions or lack any specificity for ERβ in multiple assays [47]. By using different affinity-based applications and controls, it has been subsequently demonstrated that only one of 13 antibodies tested is sufficiently specific in immunohistochemistry analysis (IHC) [48]. However, although several efforts have been made to validate the anti ERβ antibodies, a discrepancy between detectable mRNA and protein levels still remains. These controversies have been recently highlighted [49], and a brief summary of pitfalls related to ERβ antibodies is reported below. Firstly, most of the used antibodies do not show specificity. Again, IHC analysis of steroid receptors (SRs), and hence ERβ depends on the cell permeabilization (i.e., Triton X-100 or Nonidet P40), which might allow the nuclear or cytoplasmic staining of the receptors. Further, the antibodies produced against the N-terminal domain of ERβ expressed in *E. coli* work fine in recognizing the ERβ in *E. coli*, but they do not recognize the receptor in IHC analysis of tissues, because of post-transcriptional modifications in the N-terminal domain of endogenous ERβ. Finally, the antibodies raised against the C-terminal domain of ERβ do not enable the identification of ERβ modifications at the C-terminal domain. As such, the inability to specifically detect ERβ variants has obviously hindered the study of their functional role in PC. To overcome these challenges, the chicken polyclonal ERβ 503 antibodies have been proposed as the more appropriate tool, as they seem very specific for ERβ, do not cross-react with ERα, and specifically work in a wide range of fixed or frozen mouse, rat, monkey as well as human tissues. These antibodies detect ERβ expressed in prostate, breast, gastrointestinal tract, lung, immune cells, and some cells of the nervous system [49]. 

In addition to these problems, ERβ splice variants, such as ER2cx, do not bind estrogens or other available agonists [49]. As such, the majority of studies are focused on ERβ1 variant, while the functions of ERβ 2, 3, 4, and 5 variants are undervalued. Further, cellular responses activated by ERα or β might depend on a plethora of factors, including the expression of orphan nuclear receptors [50], AR [27,45], or epidermal growth factor-receptor (EGF-R). Additionally, the cross-talk between the ERα and β [51,52,53] and the intersection between ERβ [27,43] or ERα [27,54,55] with AR might control the outcome of PC- or breast (BC)- derived cells. Lastly, the ligand concentration, the ratio between the two ER isoforms and their intracellular distribution, the presence of endogenous inhibitors and the availability of transcriptional co-regulators might differentially activate ERα- or β and their mediated responses in target cells [42,56]. From these and other findings [41,57,58,59], it appears that additional studies are needed to depict a clear picture of the role of ERα or β in PC progression and EMT.

## 4. Estrogen Receptor (ER) α in Epithelial Mesenchymal Transition (EMT) of Benign Prostatic Hyperplasia (BPH)

Some years ago, it was reported that estradiol levels increase within BPH tissue [60] and that ERα mediates stromal proliferation in BPH [61]. Subsequent findings showed that BPH is driven by activation of an EMT program, accompanied by both the loss of E-cadherin and the increase of vimentin in epithelial cells from BPH specimens. Such a process, however, was correlated to abnormalities in ER and transforming growth factor (TGF)-β signaling, while ERα was almost undetectable in the stroma and epithelium of BPH tissues [62].

In contrast with these latter findings, it was subsequently reported that estrogen activation of ERα induces epithelial de-differentiation and an EMT program in BPH specimens, as well as in a rat model of BPH [63]. To corroborate these findings, it was also demonstrated by the same group that the ERα-specific agonist, PPT promotes the expression of EMT markers in benign epithelial BPH-1 and RWPE-1 cell lines, cultured in both 2D and 3D models. The ERβ-specific agonist, DPN inhibited the expression of EMT markers, and the anti-estrogen, ICI182, 780 blocks EMT and cell phenotypic switching induced by estradiol [64]. In summary, these findings pointed to the role of ERα in EMT and BPH pathogenesis. 

The discrepancies reported in the literature might be related to the different experimental conditions used (a murine model of BPH, in vitro 2D and 3D models, and BPH human specimens). As discussed above, the different antibodies used in these studies might also account for the observed controversial results.

## 5. Estrogen Receptor (ER) α in Epithelial Mesenchymal Transition (EMT) of Prostate Cancer (PC)

An increasing body of evidence has linked activation of ER to an EMT program and metastatic spreading of PC. Some of these findings are reminiscent of results reported in breast cancer (BC), where ER down-regulates E-cadherin levels and fosters EMT process [65,66]. It is noteworthy that recent studies also reported a role for the nuclear orphan estrogen-related receptor alpha (ERR) in EMT and progression of lung [67] as well as endometrial [68] cancer cells.

Estrogens down-regulate E-cadherin and up-regulate Snail and vimentin, thereby promoting EMT, cell migration, and anchorage-independent growth in human PC-derived PacMetUT1 cells. Silencing experiments together with administration of the antagonist ICI 182,780 abrogate these effects, thereby further corroborating a role for ER in EMT and invasiveness. Noticeably, ERα knockdown or its inhibition by ICI 182,780 impairs osteoblastic lesions and lung metastasis formation in a preclinical model of PC, indicating that ERα activation is required for bone and lung metastases [69]. Additionally, ERα activation by estradiol increases the invasiveness, and likely EMT, in the PC-derived VCAP cell line. Such a mechanism has been attributed to ERα-mediated transcriptional regulation of the nuclear enriched abundant transcript 1 (NEAT1), the most significantly over-expressed long non-coding RNA (lncRNA) associated with PC progression [70]. 

The role of ERα has been further corroborated by recent findings showing that estrogen activation of ERα drives the expression of neurogenic locus notch homolog protein 1 (NOTCH1), thereby enhancing stemness and the EMT phenotype in a PC mouse model. These effects ultimately lead to metastatic spreading, which in-turn is inhibited by tamoxifen treatment [71]. Altogether, these findings highlight the diagnostic, predictive, and therapeutic value of ERα in PC. 

Clinical evidence indicates, indeed, that higher-Gleason’s stage carcinomas had increased ERα expression and decreased ERβ expression [72]. Again, expression of ERα has been detected in biopsies from patients with a high PC Gleason’s score [73]. Finally, low expression of ERα in PC biopsies has been correlated with a better prognosis in patients older than 50 years old [74]. Altogether, the results discussed so far coincide with ERα oncogenic functions and indicate that ERα plays an important role in PC progression, beyond AR. Thus, ERα represents a ‘druggable’ target in patients with PC. Its targeting has been exploited in various trials with different results, as follows. Administration of high-dose tamoxifen or toremifene did not exert significant effects in patients with advanced PC [75,76] or in men with a high-grade prostatic intraepithelial neoplasia [77]. Anti-estrogens, however, also entered clinical trials to verify their efficacy in PC prevention [78,79]. A multicenter trial showed that tamoxifen reduces gynecomastia and breast pain in PC patients receiving androgen deprivation therapy (ADT) [80]. Thereafter, a phase III clinical trial with the selective estrogen receptor modulator (SERM), toremifene, in combination with ADT has shown a relative beneficial effect and a significant reduction of new vertebral fractures in patients with advanced PC [81,82,83]. 

Nowadays, SERMs have entered many clinical trials and they have been used alone, or in combination with ADT in various prostatic diseases with rather disappointing results [84]. Advances in molecular and genetic screenings, together with analysis of circulating tumor cells (CTCs) in liquid biopsies, would improve our understanding of PC molecular signatures and predict the efficacy of anti-estrogen treatments in patients with advanced disease.

## 6. Estrogen Receptor (ER) β in Epithelial Mesenchymal Transition (EMT) of Prostate Cancer (PC) 

Five splice variants of ERβ (ERβ1, ERβ2, ERβ3, ERβ4, and ERβ5) have been identified in humans. Expression of ERβ3 is limited to the testis, whereas ERβ4 can be detected at highest levels in testis, and at lower levels in spleen, thymus, ovary, mammary gland, and uterus. ERβ1, ERβ2, and ERβ5 can be found in normal prostate tissue, PC-derived cells as well as other human cancer cell lines. Neither ERβ3, nor ERβ4 are expressed in human-derived cancer cells, including PC cells [40,85]. Additionally, ERβ1 and ERβ2 isoforms are differentially expressed during the prostate cell cycle [86]. Notably, ERβ1 appears the only fully functional isoform of the ERβ family, since it forms homo-dimers and recruits co-regulators [87]. ERβ1, however, might also dimerize with androgen receptor (AR), thus regulating the transcription of AR-related genes. The most accepted view proposes that ERβ1 is a ‘gatekeeper’ of the epithelial phenotype and represents a repressor of invasion and metastasis [73]. In contrast, other ERβ variants neither form homo-dimers, nor recruit co-regulators. They, instead, represent the variable dimer partners of ERβ1 and modulate its activity [87]. Thus, the intracellular activity of ERβ may depend on ERβ1 expression and the ERβ variants ratio.

The majority of findings concerning the function of the ERβ splice variants derive from studies on ERβ1, and much evidence has reported an inverse relationship between the ERβ1 expression levels and the progression to highly aggressive and poorly differentiated PC [85,88,89,90]. Again, a significant loss of ERβ1, likely related to the decreased levels of available androgens after androgen deprivation therapy (ADT), has been observed in castration-resistant prostate cancer (CRPC) [91]. Further, ERβ1 activation by its specific ligand, 5α-androstane-3β (17β-diol), maintains the epithelial phenotype by inducing the expression of E-cadherin and inhibits migration of the androgen-independent and ERα-negative DU145 cells [92]. PC tissue staining revealed that ERβ1 expression inversely correlates with the PC progression to a high Gleason’s grade and that E-cadherin expression directly correlates with ERβ1 expression [93]. Additionally, experiments with PC3 cells xenografted in nude mice have shown that 3β-diol reduces the growth of established tumors and counteracts metastasis formation [94]. Raloxifene and tamoxifen promote the adhesion of DU145 and PC3 on laminin or fibronectin, and inhibit migration of these cells. The anti-estrogen ICI 182,780 reverses these effects, suggesting that they are specifically mediated by ERβ1, the only ER isoform expressed in DU145 and PC3 cell lines [95]. In the same cells, ormeloxifene, a clinically approved selective estrogen receptor modulator (SERM) inhibits EMT by repressing N-cadherin, Slug, Snail, vimentin and metalloproteases expression. As such, ormeloxifene reduces migration and invasiveness of PC3 and DU145 PC cells [96]. 

Taken together, these results support the concept that ERβ1 represents a favorable prognostic factor in PC and that its loss correlates with disease progression. 

ERβ1 loss is likely due to a “hypoxic” phenotype of PC cells from patients with poor prognosis, which might enable the selection of more aggressive cancer cells [97]. Mak and Colleagues have proposed a complex relationship between ERβ1, vascular endothelial growth factor (VEGF), hypoxia, and EMT [93]. Firstly, hypoxia-induced factor 1 (HIF-1α) and its target gene, the VEGF-A, two key factors of EMT, are frequently over-expressed in PC exhibiting poor prognosis. Then, a reduction in ERβ1 expression level is observed during PC progression, and hypoxia is partially responsible for this decrease [93]. In this experimental setting, the loss of ERβ1 correlates with metastatic spreading. ERβ1, indeed, represses transcription of the HIF-1α gene and promotes its degradation by proteasome. In such a way, ERβ1 would hinder EMT. Further evidence has shown that ERβ1 up-regulates propyl hydroxylase domain 2 (PHD2/EGLN1), thereby promoting hydroxylation and degradation of HIF-1α [98]. In summary, hypoxia emerges as one mechanism that facilitates the acquisition of mesenchymal features in PC cells by suppressing ERβ1 and stimulating the expression of VEGF-A mediated by HIF-1α.

Despite these findings, however, high ERβ1 protein levels have been observed in metastatic lesions in patients with recurrent PC [85,99,100] and high ERβ1 expression levels are associated with EMT in PC cells [101]. Activation of ERβ leads to EMT and migration in cells established from patients with aggressive PC [45]. Again, ERβ1 and AR might form a complex that is able to control gene expression upon estradiol treatment. Consistent with these findings, it has been shown that this complex up-regulates the expression of sex determining region Y-box (SOX4) in several PC-derived cells challenged with estradiol. Its silencing attenuates the estradiol effect on migration, invasion, and EMT [102]. Noticeably, SOX4 is a transcription factor that controls the EMT of PC cells [103] and is significantly expressed in CPRC specimens [102]. Again, a functional cooperation between hypoxia, ERβ1, and eNOS has been reported. In addition to the hypoxic phenotype, specimens from PC patients with poor prognosis exhibit an increase in NO production. In cultured cells derived from these PC specimens, challenging with estrogens and hypoxia stimuli leads to formation of ERβ1/eNOS, ERβ1/HIF-1α, or ERβ1/HIF-2α complexes. Such complexes allow chromatin remodeling and induce TERT, an early marker of PC development. Tissue microarray analysis also revealed that expression of endothelial nitric oxide synthase (eNOS) and ERβ1 or nuclear eNOS and HIF-2α represent the most relevant indicators of poor clinical outcome in PC patients [101]. Thus, in contrast with the findings previously discussed in this section, ERβ1 expression can be linked to a poor PC prognosis, further corroborating the concept that the functions of ER in PC still remain controversial. Conflicting data so far collected on the functions of ER in PC might result from the differential expression profiles of AR, ERα, and ERβ shown by PC cell lines used for in vitro studies. Additional factors (genetic and phenotypic instability, absence of cell line authentication, and the number of cell line passages in culture) might significantly influence the experimental results. Again, because of the absence of tumor stroma and/or extracellular matrix, experimental models often used to study PC progression cannot fully reproduce the dynamic events occurring *in vivo* during EMT and metastatic spreading. 

As before discussed, the two splice variants, ERβ2 (also known as ERβcx) and ERβ5 are relevant in PC as well as other human cancers [39,49,85]. They have truncated C-terminal regions, resulting in the loss of activation function 2 (AF-2) domain, and exhibit differences in the ligand binding domain (LBD) [46,104]. ERβ2 and ERβ5 variants cannot homo-dimerize, but they form heterodimers with ERβ1 upon estrogens stimulation [87]. Both ERβ2 and ERβ5 exhibit oncogenic properties and might up-regulate effectors of the metastatic process [46,87]. They are, indeed, expressed in the late stage of metastatic PC, pointing to their role in PC progression and likely EMT. Additionally, ERβ2 and ERβ5 have prognostic value in PC progression and are related to poor clinical outcomes [46]. In a Kaplan-Meier analysis, the combined expression of both nuclear ERβ2 (ERβ2 is commonly localized in the cytoplasm) and cytoplasmic ERβ5 identifies a group of patients with the shortest post-operative metastasis-free and overall survival. Again, stable ectopic expression of ERβ2 or ERβ5 enhances PC-3 cell invasiveness, while only ERβ5 is able to induce cell migration. These findings suggest that ERβ2 and ERβ5 engage different pathways to control motility or invasiveness of PC cells [46]. Whatever the mechanism, it is evident from these findings that ERβ2 and ERβ5 exert an oncogenic effect. That way, they might counteract the action of ERβ1. ERβ2 and ERβ5 interact with and stabilize HIF-1α, allowing the expression of hypoxic genes in PC [103]. Additionally, ERβ2 increases the expression of Twist1 and Slug. Such an effect correlates with a high Gleason’s score, invasiveness, and poor PC prognosis [105]. Other evidence supports the concept that ERβ1 and ERβ2 play opposite roles in PC invasiveness and EMT. PC cells often metastasize to bone and evidence suggests that ERβ1 and ERβ2 suppresses and induces the expression of the bone metastasis regulator Runt-related transcription factor 2 (RUNX2), respectively [106]. 

To date, many findings regarding the role of ER in EMT and PC progression have been reported in cultured cells. As such, their suitability in PC diagnosis is still limited, likely because of the relative homogeneity of PC cell lines, as compared with the heterogeneity of PC cells and tumor microenvironment. EMT does not seem to be a homogeneous program in cancer, but rather a spectrum of intermediate states [12]. Thus, new models reflecting the complexity of EMT *program in vivo* should be exploited to shed light into pending questions. 

## 7. Conclusions 

In recent years, significant advances in diagnosis, follow-up, and therapy of prostate cancer (PC) patients have been reached. Despite these efforts, PC often progresses towards the castration-resitant prostate cancer (CRPC) stage. Few therapeutic options are available to improve clinical outcome of patients with advanced disease and the challenge remains of how to apply targeted therapies, either in combination or in sequence approaches to achieve clinically meaningful outcome in PC patients with advanced disease. 

Although it is largely accepted that the estrogen receptors (ERs) directly or indirectly control epithelial-mesenchymal transition (EMT) and PC progression, the molecular events underlying the role of estrogens and their cognate receptors in PC progression still remain a challenge. Emerging findings render the picture more complex, and often generate more questions than they answer. When, for instance, ERα and ERβ are co-expressed in a tissue or tumor, as often occurs in PC, the formation of a heterodimer will likely yield a different transcriptional profile from that obtained if homodimers are generated in the presence of ligands. In this way, many components of the neurogenic locus notch homolog protein (Notch) signaling pathway can be differentially spliced by ERβ in breast cancer (BC) cells, through ERα/ERβ heterodimers [107]. Given the role of Notch pathway in EMT of PC [36], a similar mechanism might occur in PC to foster EMT and spreading. Additionally, the previously described cross talk between ER (α or β) and growth factor receptors (GFR) [43,108] might promote an activator loop of GFR-dependent signaling that amplifies the PI3-K and MAPK-dependent pathways and fosters EMT in PC cells. 

Figure 2 depicts the currently accepted mechanism(s) leading to EMT through activation of ERα or ERβ variants in PC cells. Studies of PC therapeutic targeting led to very promising therapies, including second-generation anti-androgens, cabazitaxel, and Radium-223 [109]. Despite these efforts, resistance to these drugs continues to develop. Further mechanistic insights into the role of estrogen/ERs axis in EMT would enhance our understanding of PC progression and allow the identification of new targets for therapeutic manipulation of this disease. 

In this era of personalized medicine, a number of drugs targeting EMT regulatory components have been used in clinical trials [110]. These compounds might be tested in combination with new ligands or antagonists of ERs to target advanced PC or make the disease more susceptible to treatment regimes.

## Figures and Tables

**Figure 1 cancers-11-01418-f001:**
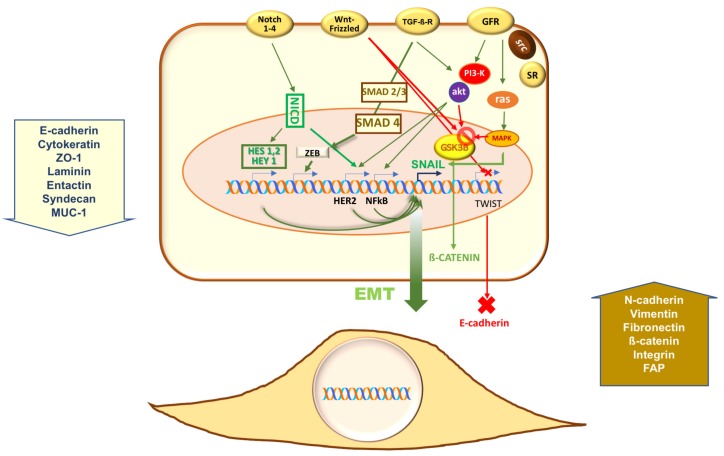
**Pathways controlling the epithelial-mesenchymal transition (EMT) program in prostate cancer (PC).** Growth factors (e.g., transforming growth factor β (TGF-β), fibroblast growth factor (FGF), hepatocyte-derived growth factor (HGF), epidermal growth factor (EGF) and insulin-like growth factor 1 (IGF-1) released by the tumor microenvironment activate their cognate receptors (GFR, growth factor receptors; TGFβ-R, TGFβ receptor) thereby triggering downstream pathways. Through phosphoinositide-3 kinase (PI3-K), mitogen activated protein kinase (MAPK), small mother against decapentaplegic (Smad), glycogen synthase kinase-3 β (GSK3β), and nuclear factor kappa-light-chain enhancer of activated B cells (NF-κB), growth factors increase the activity of transcriptional repressors of Zinc finger E-box-binding homeobox 2 (ZEB2), Twist, and Snail families. They, in turn, down-regulate E-cadherin and other epithelial cell markers, while inducing mesenchymal proteins [9,10,11,12,13,14,25,26,35]. As the ligand-bound steroid receptors (SRs) trigger rapid activation of these signaling circuits [27,28,29], it might be hypothesized that such activation also contributes to EMT in PC. Canonical wingless-INT (Wnt) is implied in EMT of PC through activation of GSK3β. Stabilization and nuclear translocation of β-catenin then follow. This latter event leads to phosphorylation and nuclear translocation of snail family transcriptional repressors 1 (SNAI1). In such a way, the transcriptional activity of SNAI1 is enhanced [35]. Neurogenic locus notch homolog protein (Notch) 1–4 trans-membrane receptors can be activated by membrane-tethered ligands in mammals. Proteolytic cleavage of Notch receptors then follows, with the subsequent release of the Notch intracellular domain (NICD), which then enter nuclei to activate the transcription factors hairy and enhancer of split (HES) 1 and 2 as well as HES 1 related (HEY1). Through NF-kB, they control the expression of genes involved in EMT, such as Snail [13]. Mutually reinforcing mechanisms promoting the EMT program also occur. HES 1 and 2 as well as HEY1, indeed, positively affect HER2 (Erb-B2) transcription, thereby creating an activator loop of Erb-B2-dependent signaling. In such a way, PI3-K- and MAPK-dependent signaling is amplified. A role for the Notch pathway in EMT and spreading of PC has been reported [36].

**Figure 2 cancers-11-01418-f002:**
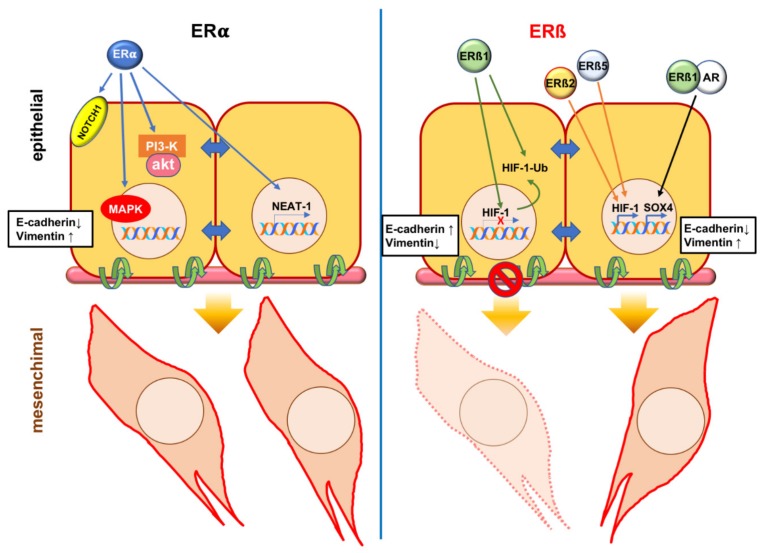
**Estrogen receptor (ER) α or β control the epithelial mesenchymal transition (EMT) program in prostate cancer (PC). Left panel**: the ligand activated ERα might induce EMT through transcriptional up-regulation of Neurogenic locus notch homolog protein 1 (NOTCH1) [71] or nuclear enriched abundant transcript 1 (NEAT-1) [70]. Phosphoinositide 3-kinase (PI3-K)/AKT and Ras/Mitogen activated protein kinase (MAPK) pathways both induce EMT of PC [25,26]. As the ligand-bound ERα triggers activation of both these pathways in target cells [27,28,29], it might be argued that such activation leads to EMT in PC. **Right panel**: the ligand activated ERβ1 hinders EMT by repressing the transcription of hypoxia-induced factor 1α (HIF-1α) and promoting its degradation by proteasome [93]. By stabilizing HIF-1α, ERβ2 and 5 promote EMT [46]. Under certain conditions, ERβ1 could dimerize with androgen receptor (AR) upon being challenged with estrogens. Sex determining region Y box 4 (SOX4) up-regulation follows and this event leads to EMT in PC [102].

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
