# Peer review of "Estrogen Receptors in Epithelial-Mesenchymal Transition of Prostate Cancer"

_cancers, 2019, doi:10.3390/cancers11101418_

Round 1

Reviewer 1 Report

The review "Estrogen receptors in epithelial-mesenchymal transition of prostate cancer" by Di Zazzo et al. is describing what is known about estrogen receptors and effect on EMT in prostate cancer. Although the knowledge of estrogen receptor function in prostate is limited and so far no transcriptome affected by estrogen receptors have been identified in prostate cell lines. The review is infomative and reflects the current stage of knowledge.

On page 5 line 203-204 is described 5 splice variants of ERbeta. However it is still unclear if ERbeta3 and ERbeta4 is expressed in prostate or prostate cell lines. This should be made clear not to cause confusion.

On page 7 line 306 there is a misspelling SOXO4 should be SOX4

Author Response

‘The review "Estrogen receptors in epithelial-mesenchymal transition of prostate cancer" by Di Zazzo et al. is describing what is known about estrogen receptors and effect on EMT in prostate cancer. Although the knowledge of estrogen receptor function in prostate is limited and so far no transcriptome affected by estrogen receptors have been identified in prostate cell lines. The review is infomative and reflects the current stage of knowledge.’

- We thank the Reviewer for these encouraging comments. The revised version of the manuscript , however, has been further improved by discussion of findings concerning the molecular mechanisms underlying the EMT in cancer cells, including PC cells. Therefore, the manuscript now contains an additional Figure illustrating the most important pathways involved in EMT (see the new Figure 1). Additionally, some sections have been splitted and re-organized to make more easy the reading of the manuscript.

‘On page 5 line 203-204 is described 5 splice variants of ERbeta. However it is still unclear if ERbeta3 and ERbeta4 is expressed in prostate or prostate cell lines. This should be made clear not to cause confusion’.

- We completely agree with the reviewer’s comment. Therefore, we have extensively  clarified this issue (see page 9, lines 274-286 and page 11, lines 347-354 in the revised manuscript).

‘On page 7 line 306 there is a misspelling SOXO’

The misspelling has been corrected, together with many typos errors.

All the changes in the revised manuscript are highlighted in yellow.

Reviewer 2 Report

The review by Zazzo et al. synthesizes the latest discoveries on the estrogen signaling pathway in prostate cancer, and more specifically the role of the estrogen receptors in the EMT transition.They review an important point that needs to be addressed in the prostate cancer research community. Indeed, as a male disease, it is often seen as the target of the androgen signaling pathway. However, and as clearly described by the authors, the classic female hormones, estrogens, are also key players in prostate cancer progression. Yet, before being acceptable for publication, several concerns need to be addressed.

1. Given that the review focuses on the effect of ERs on the EMT transition, a graphical model of EMT and a better molecular description of this process would add value to the current review. It would also help highlight the different molecular functions of ERs on this biological pathway in the current figure of the review. For example, on lane 178, the authors mention that ERa controls NOTCH1, which enhances stemness and EMT phenotype. However, at this stage of the review, NOTCH signaling was never mentioned nor described. Yet, NOTCH1 seems important, and it is in the review figure, yet its function is rather still undetermined even in their model. This should be clearly described in the revised version of the manuscript, adding a figure model of EMT and enhancing their actual figure.

2. Section 4 of the review is "ERa in EMT in PC"; however, the first half is actually ERa in EMT in BPH and in non-tumor prostate cells. Given the discrepancies in the literature, as noted by the authors, clearly separating this section in 2 (BPH vs PC) might allow a better appreciation of ERa regulation of EMT.

3. The review is well written and structured. However, due to the paradox effect of ERb and its variants in PCa, this latter section is confusing. As an example, lanes 220-221, "... inverse relationship between the ERb expression levels and the progression of highly aggressive and poorly differentiated PC." vs lanes 265-266 "... high ERb protein levels have been observed in metastatic lesions in patients with recurrent PC.".  It think that what is adding confusion is the jumps that are made between the different molecular mechanisms. In between the two opposing sentences in lanes 220-221 and 265-266, the authors discuss about the ERb variants, interactions with HIF1a, etc. The authors should re-structure this section to have a more logical order, such as discussing all the ERb variant 1 mecanisms, which possibly changes during PC evolution, its interactions with other factors, and then with ERb variants 2-5. 

4. Lines 128-130: the authors cite 2 articles and mention that ERbeta available antibodies do not easily allow the identification of the different receptor variants. The two articles they refer too actually demonstrate that MOST ERb antibodies are NOT specific at all, neither for ERb1 or other variants. This should be clearly stated and thus the text must be changed to better reflect the conclusions of these 2 key articles. It is also not clear how these unspecific antibodies have added confusion in our understanding of the ER functions in PCa, maybe the authors could better addressed this particular point.

Author Response

‘The review by Zazzo et al. synthesizes the latest discoveries on the estrogen signaling pathway in prostate cancer, and more specifically the role of the estrogen receptors in the EMT transition. They review an important point that needs to be addressed in the prostate cancer research community…………..Yet, before being acceptable for publication, several concerns need to be addressed’.

- We very much appreciated the comments raised by the Referee, therefore we have improved our manuscript according to the comments indicated below.

‘Given that the review focuses on the effect of ERs on the EMT transition, a graphical model of EMT and a better molecular description of this process would add value to the current review. It would also help highlight the different molecular functions of ERs on this biological pathway in the current figure of the review. For example, on lane 178, the authors mention that ERa controls NOTCH1, which enhances stemness and EMT phenotype. However, at this stage of the review, NOTCH signaling was never mentioned nor described. Yet, NOTCH1 seems important, and it is in the review figure, yet its function is rather still undetermined even in their model. This should be clearly described in the revised version of the manuscript, adding a figure model of EMT and enhancing their actual figure’.

We agree with the Referee’s comments. Therefore, we have now added a new Figure (Figure 1), which depicts the ‘state of art’of the current accepted molecular mechanisms underlying the EMT process in human prostate cancers. Details of these pathway have been discussed on page                , lines    of the revised manuscript. The role of NOTCH signaling has been discussed, together with its impact in EMT of PC (see page 4, lines 116-133 and page 5, lines 134-155 in the revised manuscript).

Section 4 of the review is "ERa in EMT in PC"; however, the first half is actually ERa in EMT in BPH and in non-tumor prostate cells. Given the discrepancies in the literature, as noted by the authors, clearly separating this section in 2 (BPH vs PC) might allow a better appreciation of ERa regulation of EMT.

Consistent with the Referee’s comments, we have now splitted the section 4 in two different sections. The first (section 4) focuses on BPH, the second (section 5) on PC.

The review is well written and structured. However, due to the paradox effect of ERb and its variants in PCa, this latter section is confusing. As an example, lanes 220-221, "... inverse relationship between the ERb expression levels and the progression of highly aggressive and             poorly differentiated PC." vs lanes 265-266 "... high ERb protein levels have been observed   in metastatic lesions in patients with recurrent PC.".  It think that what is adding confusion      is the jumps that are made between the different molecular mechanisms. In between the     two opposing sentences in lanes 220-221 and 265-266, the authors discuss about the ERb   variants, interactions with HIF1a, etc. The authors should re-structure this section to have a             more logical order, such as discussing all the ERb variant 1 mechanisms, which possibly         changes during PC evolution, its interactions with other factors, and then with ERb variants      2-5. 

- We very much appreciated the suggestion of the Referee. Therefore we have completely changed the organization and structure of this section (section 6 in the revised manuscript).

Lines 128-130: the authors cite 2 articles and mention that ERbeta available antibodies do not easily allow the identification of the different receptor variants. The two articles they refer too actually demonstrate that MOST ERb antibodies are NOT specific at all, neither for ERb1 or other variants. This should be clearly stated and thus the text must be changed to better reflect the conclusions of these 2 key articles. It is also not clear how these unspecific antibodies have added confusion in our understanding of the ER functions in PCa, maybe the authors could better addressed this particular point.

- Consistent with the Referee’s comment, we have now completely revised this paragraph. Tips, tricks and pitfalls related to the use of anti ER beta antibodies have been presented and discussed, together with the obstacles encountered during the studies on the role of ER beta in PC.

In this regard see also page 6, lines 178-199  of the revised manuscript.  

All the changes in the revised manuscript are highlighted in yellow.

Reviewer 3 Report

This review has thoroughly discussed the current studies of Estrogen receptors in the EMT of Prostate cancer. The author split the discussion based on different ERs, since ER-alpha and ER-beta played opposite roles in the metastasis of prostate cancer. However, several recent studies also investigated an interaction between ER-alpha and ER-beta and its impact on mRNA splicing machinery. It would be more thorough if the author can look into these studies and discuss whether this interaction between ER-alpha and ER-beta also impacts on the invasiveness of prostate cancer.

Overall, this review is well-organized and bring insightful information to the field of prostate cancer research.

Author Response

This review has thoroughly discussed the current studies of Estrogen receptors in the EMT    of Prostate cancer. The author split the discussion based on different ERs, since ER-alpha           and ER-beta played opposite roles in the metastasis of prostate cancer. However, several      recent studies also investigated an interaction between ER-alpha and ER-beta and its         impact on mRNA splicing machinery. It would be more thorough if the author can look into   these studies and discuss whether this interaction between ER-alpha and ER-beta also          impacts on the invasiveness of prostate cancer.

            Overall, this review is well-organized and bring insightful information to the field of   prostate cancer research’.

- We thank the Referee for the encouraging comments, and we very much appreciated the concern about the role of ERalpha/beta in mRNA splicing machinery. To our knowledge, such interaction certainly impacts the Notch pathway in breast cancer cells. However, because of the role of Notch signaling in ERalpha-induced EMT and metastatic events of PC, it is very likely that a similar interaction might occur also in PC, thereby controlling EMT and invasiveness.

These findings have been introduced in the revised version of the manuscript (page 12, lines 382-388) and related Refs have been added.

All the changes in the revised manuscript are highlighted in yellow.

Round 2

Reviewer 2 Report

The authors answered to all my concerns. Congratulations for your good work.

Author Response

Dear Referee,

Many thanks for your concerns.

They have certainly made the manuscript more exhaustive and clear.

Best regards,

Gabriella Castoria